# Multiple-Play Bandits in the Position-Based Model

**Paul Lagrée**[*]
LRI, Université Paris Sud
Université Paris Saclay
paul.lagree@u-psud.fr

**Claire Vernade**[*]
LTCI, CNRS, Télécom ParisTech
Université Paris Saclay
vernade@enst.fr

**Olivier Cappé**
LTCI, CNRS
Télécom ParisTech
Université Paris Saclay

## Abstract

Sequentially learning to place items in multi-position displays or lists is a task that can be cast into the multiple-play semi-bandit setting. However, a major concern in this context is when the system cannot decide whether the user feedback for each item is actually exploitable. Indeed, much of the content may have been simply ignored by the user. The present work proposes to exploit available information regarding the display position bias under the so-called Position-based click model (PBM). We first discuss how this model differs from the Cascade model and its variants considered in several recent works on multiple-play bandits. We then provide a novel regret lower bound for this model as well as computationally efficient algorithms that display good empirical and theoretical performance.

## 1  Introduction

During their browsing experience, users are constantly provided – without having asked for it – with clickable content spread over web pages. While users interact on a website, they send clicks to the system for a very limited selection of the clickable content. Hence, they let every unclicked item with an equivocal answer: the system does not know whether the content was really deemed irrelevant or simply ignored. In contrast, in traditional multi-armed bandit (MAB) models, the learner makes actions and observes at each round the reward corresponding to the chosen action. In the so-called multiple play semi-bandit setting, when users are presented with $L$ items, they are assumed to provide feedback for each of those items.

Several variants of this basic setting have been considered in the bandit literature. The necessity for the user to provide feedback for each item has been called into question in the context of the so-called *Cascade Model* [8, 14, 6] and its extensions such as the *Dependent Click Model* (DCM) [20]. Both models are particularly suited for search contexts, where the user is assumed to be looking for something relative to a query. Consequently, the learner expects explicit feedback: in the Cascade Model each valid observation sequence must be either all zeros or terminated by a one, such that no ambiguity is left on the evaluation of the presented items, while multiple clicks are allowed in the DCM thus leaving some ambiguity on the last zeros of a sequence.

In the Cascade Model, the positions of the items are not taken into account in the reward process because the learner is assumed to obtain a click as long as the interesting item belongs to the list. Indeed, there are even clear indications that the optimal strategy in a learning context consists in showing the most relevant items at the end of the list in order to maximize the amount of observed feedback [14] – which is counter-intuitive in recommendation tasks.

To overcome these limitations, [6] introduces weights – to be defined by the learner – that are attributed to positions in the list, with a click on position $l \in \{1, \ldots, L\}$ providing a reward $w_l$, where the sequence $(w_l)_l$ is decreasing to enforce the ranking behavior. However, no rule is given for

---

[*]The two authors contributed equally.

setting the weights $(w_l)_l$ that control the order of importance of the positions. The authors propose an algorithm based on KL-UCB [10] and prove a lower bound on the regret as well as an asymptotically optimal upper bound.

Another way to address the limitations of the Cascade Model is to consider the DCM as in [20]. Here, examination probabilities $v_l$ are introduced for each position $l$: conditionally on the event that the user effectively scanned the list up to position $l$, he/she can choose to leave with probability $v_l$ and in that case, the learner is aware of his/her departure. This framework naturally induces the necessity to rank the items in the optimal order.

All previous models assume that a portion of the recommendation list is explicitly examined by the user and hence that the learning algorithm eventually has access to rewards corresponding to the unbiased user's evaluation of each item. In contrast, we propose to analyze multiple-play bandits in the Position-based model (PBM) [5]. In the PBM, each position in the list is also endowed with a binary *Examination variable* [8, 19] which is equal to one only when the user paid attention to the corresponding item. But this variable, that is independent of the user's evaluation of the item, is not observable. It allows to model situations where the user is not explicitly looking for specific content, as in typical recommendation scenarios.

Compared to variants of the Cascade model, the PBM is challenging due to the censoring induced by the examination variables: the learning algorithm observes actual clicks but non-clicks are always ambiguous. Thus, combining observations made at different positions becomes a non-trivial statistical task. Some preliminary ideas on how to address this issue appear in the supplementary material of [13]. In this work, we provide a complete statistical study of stochastic multiple-play bandits with semi-bandit feedback in the PBM.

We introduce the model and notations in Section 2 and provide the lower bound on the regret in Section 3. In Section 4, we present two optimistic algorithms as well as a theoretical analysis of their regret. In the last section dedicated to experiments, those policies are compared to several benchmarks on both synthetic and realistic data.

## 2 Setting and Parameter Estimation

We consider the binary stochastic bandit model with $K$ Bernoulli-distributed arms. The model parameters are the arm expectations $\theta = (\theta_1, \theta_2, \ldots, \theta_K)$, which lie in $\Theta = (0,1)^K$. We will denote by $\mathcal{B}(\theta)$ the Bernoulli distribution with parameter $\theta$ and by $d(p,q) := p \log(p/q) + (1 - p) \log((1-p)/(1-q))$ the Kullback-Leibler divergence from $\mathcal{B}(p)$ to $\mathcal{B}(q)$. At each round $t$, the learner selects a list of $L$ arms – referred to as an *action* – chosen among the $K$ arms which are indexed by $k \in \{1, \ldots, K\}$. The set of actions is denoted by $\mathcal{A}$ and thus contains $K!/(K - L)!$ ordered lists; the action selected at time $t$ will be denoted $A(t) = (A_1(t), \ldots, A_L(t))$.

The PBM is characterized by examination parameters $(\kappa_l)_{1 \le l \le L}$, where $\kappa_l$ is the probability that the user effectively observes the item in position $l$ [5]. At round $t$, the selection $A(t)$ is shown to the user and the learner observes the complete feedback – as in semi-bandit models – but the observation at position $l$, $Z_l(t)$, is *censored* being the product of two independent Bernoulli variables $Y_l(t)$ and $X_l(t)$, where $Y_l(t) \sim \mathcal{B}(\kappa_l)$ is non null when the user considered the item in position $l$ – which is unknown to the learner – and $X_l(t) \sim \mathcal{B}(\theta_{A_l(t)})$ represents the actual user feedback to the item shown in position $l$. The learner receives a reward $r_{A(t)} = \sum_{l=1}^{L} Z_l(t)$, where $Z(t) = (X_1(t)Y_1(t), \ldots, X_L(t)Y_L(t))$ denotes the vector of censored observations at step $t$.

In the following, we will assume, without loss of generality, that $\theta_1 > \cdots > \theta_K$ and $\kappa_1 > \cdots > \kappa_L > 0$, in order to simplify the notations. The fact that the sequences $(\theta_l)_l$ and $(\kappa_l)_l$ are decreasing implies that the optimal list is $a^* = (1, \ldots, L)$. Denoting by $R(T) = \sum_{t=1}^{T} r_{a^*} - r_{A(t)}$ the regret incurred by the learner up to time $T$, one has

$$\mathbb{E}[R(T)] = \sum_{t=1}^{T} \sum_{l=1}^{L} \kappa_l (\theta_{a_l^*} - \mathbb{E}[\theta_{A_l(t)}]) = \sum_{a \in \mathcal{A}} (\mu^* - \mu_a) \mathbb{E}[N_a(T)] = \sum_{a \in \mathcal{A}} \Delta_a \mathbb{E}[N_a(T)], \quad (1)$$

where $\mu_a = \sum_{l=1}^{L} \kappa_l \theta_{a_l}$ is the expected reward of action $a$, $\mu^* = \mu_{a^*}$ is the best possible reward in average, $\Delta_a = \mu^* - \mu_a$ the expected gap to optimality, and, $N_a(T) = \sum_{t=1}^{T} \mathbb{1}\{A(t) = a\}$ is the number of times action $a$ has been chosen up to time $T$.

In the following, we assume that the examination parameters $(\kappa_l)_{1 \leq l \leq L}$ are known to the learner. These can be estimated from historical data [5], using, for instance, the EM algorithm [9] (see also Section 5). In most scenarios, it is realistic to assume that the content (e.g., ads in on-line advertising) is changing much more frequently than the layout (web page design for instance) making it possible to have a good knowledge of the click-through biases associated with the display positions.

The main statistical challenge associated with the PBM is that one needs to obtain estimates and confidence bounds for the components $\theta_k$ of $\theta$ from the available $\mathcal{B}(\kappa_l \theta_k)$-distributed draws corresponding to occurrences of arm $k$ at various positions $l = 1, \ldots, L$ in the list. To this aim, we define the following statistics: $S_{k,l}(t) = \sum_{s=1}^{t-1} Z_l(s) \mathbb{1}\{A_l(s) = k\}$, $S_k(t) = \sum_{l=1}^{L} S_{k,l}(t)$, $N_{k,l}(t) = \sum_{s=1}^{t-1} \mathbb{1}\{A_l(s) = k\}$, $N_k(t) = \sum_{l=1}^{L} N_{k,l}(t)$. We further require bias-corrected versions of the counts $\tilde{N}_{k,l}(t) = \sum_{s=1}^{t-1} \kappa_l \mathbb{1}\{A_l(s) = k\}$ and $\tilde{N}_k(t) = \sum_{l=1}^{L} \tilde{N}_{k,l}(t)$.

A time $t$, and *conditionally on the past actions* $A(1)$ *up to* $A(t-1)$, the Fisher information for $\theta_k$ is given by $I(\theta_k) = \sum_{l=1}^{L} N_{k,l}(t) \kappa_l / (\theta_k(1 - \kappa_l \theta_k))$ (see Appendix A). We cannot however estimate $\theta_k$ using the maximum likelihood estimator since it has no closed form expression. Interestingly though, the simple pooled linear estimator

$$\hat{\theta}_k(t) = S_k(t) / \tilde{N}_k(t), \tag{2}$$

considered in the supplementary material to [13], is unbiased and has a (conditional) variance of $\upsilon(\theta_k) = (\sum_{l=1}^{L} N_{k,l}(t) \kappa_l \theta_k (1 - \kappa_l \theta_k)) / (\sum_{l=1}^{L} N_{k,l}(t) \kappa_l)^2$, which is close to optimal given the Cramér-Rao lower bound. Indeed, $\upsilon(\theta_k) I(\theta_k)$ is recognized as a ratio of a weighted arithmetic mean to the corresponding weighted harmonic mean, which is known to be larger than one, but is upper bounded by $1/(1 - \theta_k)$, irrespectively of the values of the $\kappa_l$'s. Hence, if, for instance, we can assume that all $\theta_k$'s are smaller than one half, the loss with respect to the best unbiased estimator is no more than a factor of two for the variance. Note that despite its simplicity, $\hat{\theta}_k(t)$ cannot be written as a simple sum of conditionally independent increments divided by the number of terms and will thus require specific concentration results.

It can be checked that when $\theta_k$ gets very close to one, $\hat{\theta}_k(t)$ is no longer close to optimal. This observation also has a Bayesian counterpart that will be discussed in Section 5. Nevertheless, it is always preferable to the "position-debiased" estimator $(\sum_{l=1}^{L} S_{k,l}(t) / \kappa_l) / N_{k,l}(t)$ which gets very unreliable as soon as one of the $\kappa_l$'s gets very small.

## 3    Lower Bound on the Regret

In this section, we consider the fundamental asymptotic limits of learning performance for online algorithms under the PBM. These cannot be deduced from earlier general results, such as those of [11, 7], due to the censoring in the feedback associated to each action. We detail a simple and general proof scheme – using the results of [12] – that applies to the PBM, as well as to more general models.

Lower bounds on the regret rely on changes of measure: the question is how much can we mistake the true parameters of the problem for others, when observing successive arms? With this in mind, we will subscript all expectations and probabilities by the parameter value and indicate explicitly that the quantities $\mu_a, a^*, \mu^*, \Delta_a$, introduced in Section 2, also depend on the parameter. For ease of notation, we will still assume that $\theta$ is such that $a^*(\theta) = (1, \ldots, L)$.

### 3.1    Existing results for multiple-play bandit problems

Lower bounds on the regret will be proved for *uniformly efficient* algorithms, in the sense of [16]:

**Definition 1.** *An algorithm is said to be* uniformly efficient *if for any bandit model parameterized by $\theta$ and for all $\alpha \in (0, 1]$, its expected regret after $T$ rounds is such that $\mathbb{E}_\theta R(T) = o(T^\alpha)$.*

For the multiple-play MAB, [2] obtained the following bound

$$\liminf_{T \to \infty} \frac{\mathbb{E}_\theta R(T)}{\log(T)} \geq \sum_{k=L+1}^{K} \frac{\theta_L - \theta_k}{d(\theta_k, \theta_L)}. \tag{3}$$

For the "learning to rank" problem where rewards follow the weighted Cascade Model with decreasing weights $(w_l)_{l=1,...,L}$, [6] derived the following bound

$$\liminf_{T\to\infty} \frac{\mathbb{E}_\theta R(T)}{\log T} \geq w_L \sum_{k=L+1}^{K} \frac{\theta_L - \theta_k}{d(\theta_k, \theta_L)}.$$

Perhaps surprisingly, this lower bound does not show any additional term corresponding to the complexity of ranking the $L$ optimal arms. Indeed, the errors are still asymptotically dominated by the need to discriminate irrelevant arms $(\theta_k)_{k>L}$ from the worst of the relevant arms, that is, $\theta_L$.

### 3.2  Lower bound step by step

**Step 1: Computing the expected log-likelihood ratio.**   Denoting by $\mathcal{F}_{s-1}$ the $\sigma$-algebra generated by the past actions and observations, we define the log-likelihood ratio for the two values $\theta$ and $\lambda$ of the parameters by

$$\ell(t) := \sum_{s=1}^{t} \log \frac{p(Z(s); \theta \mid \mathcal{F}_{s-1})}{p(Z(s); \lambda \mid \mathcal{F}_{s-1})}. \tag{4}$$

**Lemma 2.**  *For each position $l$ and each item $k$, define the local amount of information by*

$$I_l(\theta_k, \lambda_k) := \mathbb{E}_\theta \left[ \log \frac{p(Z_l(t); \theta)}{p(Z_l(t); \lambda)} \,\middle|\, A_l(t) = k \right],$$

*and its cumulated sum over the $L$ positions by $I_a(\theta, \lambda) := \sum_{l=1}^{L} \sum_{k=1}^{K} \mathbb{1}\{a_l = k\} I_l(\theta_k, \lambda_k)$. The expected log-likelihood ratio is given by*

$$\mathbb{E}_\theta[\ell(t)] = \sum_{a \in \mathcal{A}} I_a(\theta, \lambda) \mathbb{E}_\theta[N_a(t)]. \tag{5}$$

The next proposition is adapted from Theorem 17 in Appendix B of [12] and provides a lower bound on the expected log-likelihood ratio.

**Proposition 3.**  *Let $B(\theta) := \{\lambda \in \Theta \mid \forall l \leq L, \theta_l = \lambda_l \text{ and } \mu^*(\theta) < \mu^*(\lambda)\}$ be the set of changes of measure that improve over $\theta$ without modifying the optimal arms. Assuming that the expectation of the log-likelihood ratio may be written as in (5), for any uniformly efficient algorithm one has*

$$\forall \lambda \in B(\theta), \quad \liminf_{T\to\infty} \frac{\sum_{a \in \mathcal{A}} I_a(\theta, \lambda) \mathbb{E}_\theta[N_a(T)]}{\log(T)} \geq 1.$$

**Step 2: Variational form of the lower bound.**   We are now ready to obtain the lower bound in a form similar to that originally given by [11].

**Theorem 4.**  *The expected regret of any uniformly efficient algorithm satisfies*

$$\liminf_{T\to\infty} \frac{\mathbb{E}_\theta R(T)}{\log T} \geq f(\theta), \quad \text{where } f(\theta) = \inf_{c \succeq 0} \sum_{a \in \mathcal{A}} \Delta_a(\theta) c_a, \quad s.t. \quad \inf_{\lambda \in B(\theta)} \sum_{a \in \mathcal{A}} I_a(\theta, \lambda) c_a \geq 1.$$

Theorem 4 is a straightforward consequence of Proposition 3, combined with the expression of the expected regret given in (1). The vector $c \in \mathbb{R}_+^{|\mathcal{A}|}$, that satisfies the inequality $\sum_{a \in \mathcal{A}} I_a(\theta, \lambda) c_a \geq 1$, represents the feasible values of $\mathbb{E}_\theta[N_a(T)] / \log(T)$.

**Step 3: Relaxing the constraints.**   The bounds mentioned in Section 3.1 may be recovered from Theorem 4 by considering only the changes of measure that affect a single suboptimal arm.

**Corollary 5.**

$$f(\theta) \geq \inf_{c \succeq 0} \sum_{a \in \mathcal{A}} \Delta_a(\theta) c_a, \quad s.t. \quad \sum_{a \in \mathcal{A}} \sum_{l=1}^{L} \mathbb{1}\{a_l = k\} I_l(\theta_k, \theta_L) c_a \geq 1, \quad \forall k \in \{L+1, \ldots, K\}.$$

Corollary 5 is obtained by restricting the constraint set $B(\theta)$ of Theorem 4 to $\cup_{k=L+1}^{K} B_k(\theta)$, where $B_k(\theta) := \{\lambda \in \Theta \mid \forall j \neq k, \theta_j = \lambda_j \text{ and } \mu^*(\theta) < \mu^*(\lambda)\}$.

### 3.3 Lower bound for the PBM

**Theorem 6.** *For the PBM, the following lower bound holds for any uniformly efficient algorithm:*

$$\liminf_{T \to \infty} \frac{\mathbb{E}_\theta R(T)}{\log T} \geq \sum_{k=L+1}^{K} \min_{l \in \{1,\dots,L\}} \frac{\Delta_{v_{k,l}}(\theta)}{d(\kappa_l \theta_k, \kappa_l \theta_L)}, \tag{6}$$

*where $v_{k,l} := (1, \dots, l-1, k, l, \dots, L-1)$.*

*Proof.* First, note that for the PBM one has $I_l(\theta_k, \lambda_k) = d(\kappa_l \theta_k, \kappa_l \lambda_k)$. To get the expression given in Theorem 6 from Corollary 5, we proceed as in [6] showing that the optimal coefficients $(c_a)_{a \in \mathcal{A}}$ can be non-zero only for the $K - L$ actions that put the suboptimal arm $k$ in the position $l$ that reaches the minimum of $\Delta_{v_{k,l}}(\theta)/d(\kappa_l \theta_k, \kappa_l \theta_L)$. Nevertheless, this position does not always coincide with $L$, the end of the displayed list, contrary to the case of [6] (see Appendix B for details). $\qquad\square$

The discrete minimization that appears in the r.h.s. of Theorem 6 corresponds to a fundamental trade-off in the PBM. When trying to discriminate a suboptimal arm $k$ from the $L$ optimal ones, it is desirable to put it higher in the list to obtain more information, as $d(\kappa_l \theta_k, \kappa_l \theta_L)$ is an increasing function of $\kappa_l$. On the other hand, the gap $\Delta_{v_{k,l}}(\theta)$ is also increasing as $l$ gets closer to the top of the list. The fact that $d(\kappa_l \theta_k, \kappa_l \theta_L)$ is not linear in $\kappa_l$ (it is a strictly convex function of $\kappa_l$) renders the trade-off non trivial. It is easily checked that when $(\theta_1 - \theta_L)$ is very small, i.e. when all optimal arms are equivalent, the optimal exploratory position is $l = 1$. In contrast, it is equal to $L$ when the gap $(\theta_L - \theta_{L+1})$ becomes very small. Note that by using that for any suboptimal $a \in \mathcal{A}$, $\Delta_a(\theta) \geq \sum_{k=L+1}^{K} \sum_{l=1}^{L} \mathbb{1}\{a_l = k\}\kappa_l(\theta_L - \theta_k)$, one can lower bound the r.h.s. of Theorem 6 by $\kappa_L \sum_{k=L+1}^{K}(\theta_L - \theta_k)/d(\kappa_L \theta_k, \kappa_L \theta_L)$, which is not tight in general.

**Remark 7.** *In the uncensored version of the PBM – i.e., if the $Y_l(t)$ were observed –, the expression of $I_a(\theta, \lambda)$ is simpler: it is equal to $\sum_{l=1}^{L} \sum_{k=1}^{K} \mathbb{1}\{A_l(t) = k\}\kappa_l d(\theta_k, \lambda_k)$ and leads to a lower bound that coincides with (3). The uncensored PBM is actually statistically very close to the weighted Cascade model and can be addressed by algorithms that do not assume knowledge of the $(\kappa_l)_l$ but only of their ordering.*

## 4 Algorithms

In this section we introduce two algorithms for the PBM. The first one uses the CUCB strategy of [4] and requires an simple upper confidence bound for $\theta_k$ based on the estimator $\hat{\theta}_k(t)$ defined in (2). The second algorithm is based on the *Parsimonious Item Exploration* – PIE(L) – scheme proposed in [6] and aims at reaching asymptotically optimal performance. For this second algorithm, termed PBM-PIE, it is also necessary to use a multi-position analog of the well-known KL-UCB index [10] that is inspired by a result of [17]. The analysis of PBM-PIE provided below confirms the relevance of the lower bound derived in Section 3.

**PBM-UCB** The first algorithm simply consists in sorting optimistic indices in decreasing order and pulling the corresponding first $L$ arms [4]. To derive the expression of the required "exploration bonus" we use an upper confidence for $\hat{\theta}_k(t)$ based on Hoeffding's inequality:

$$U_k^{UCB}(t, \delta) = \frac{S_k(t)}{\tilde{N}_k(t)} + \sqrt{\frac{N_k(t)}{\tilde{N}_k(t)}} \sqrt{\frac{\delta}{2\tilde{N}_k(t)}},$$

for which a coverage bound is given by the next proposition, proven in Appendix C.

**Proposition 8.** *Let $k$ be any arm in $\{1, \dots, K\}$, then for any $\delta > 0$,*

$$\mathbb{P}\left(U_k^{UCB}(t, \delta) \leq \theta_k\right) \leq e\delta \log(t) e^{-\delta}.$$

Following the ideas of [7], it is possible to obtain a logarithmic regret upper bound for this algorithm. The proof is given in Appendix D.

**Theorem 9.** *Let* $C(\kappa) = \min_{1 \leq l \leq L}[(\sum_{j=1}^{L} \kappa_j)^2/l + (\sum_{j=1}^{l} \kappa_j)^2]/\kappa_L^2$ *and* $\Delta = \min_{a \in \sigma(a^*) \setminus a^*} \Delta_a$, *where* $\sigma(a^*)$ *denotes the permutations of the optimal action. Using* PBM-UCB *with* $\delta = (1 + \epsilon) \log(t)$ *for some* $\epsilon > 0$, *there exists a constant* $C_0(\epsilon)$ *independent from the model parameters such that the regret of* PBM-UCB *is bounded from above by*

$$\mathbb{E}[R(T)] \leq C_0(\epsilon) + 16(1 + \epsilon)C(\kappa) \log T \left( \frac{L}{\Delta} + \sum_{k \notin a^*} \frac{1}{\kappa_L(\theta_L - \theta_k)} \right).$$

The presence of the term $L/\Delta$ in the above expression is attributable to limitations of the mathematical analysis. On the other hand, the absence of the KL-divergence terms appearing in the lower bound (6) is due to the use of an upper confidence bound based on Hoeffding's inequality.

**PBM-PIE** We adapt the PIE($l$) algorithm introduced by [6] for the Cascade Model to the PBM in Algorithm 1 below. At each round, the learner potentially explores at position $L$ with probability $1/2$ using the following upper-confidence bound for each arm $k$

$$U_k(t, \delta) = \sup_{q \in [\theta_k^{\min}(t), 1]} \left\{ q \left| \sum_{l=1}^{L} N_{k,l}(t) d \left( \frac{S_{k,l}(t)}{N_{k,l}(t)}, \kappa_l q \right) \leq \delta \right. \right\}, \tag{7}$$

where $\theta_k^{\min}(t)$ is the minimum of the convex function $\Phi : q \mapsto \sum_{l=1}^{L} N_{k,l}(t) d(S_{k,l}(t)/N_{k,l}(t), \kappa_l q)$. In other positions, $l = 1, \ldots, L - 1$, PBM-PIE selects the arms with the largest estimates $\hat{\theta}_k(t)$. The resulting algorithm is presented as Algorithm 1 below, denoting by $\mathcal{L}(t)$ the $L$-largest empirical estimates, referred to as the "leaders" at round $t$.

---

**Algorithm 1 – PBM-PIE**

---

**Require:** $K, L$, observation probabilities $\kappa, \epsilon > 0$
  Initialization: first $K$ rounds, play each arm at every position
  **for** $t = K + 1, \ldots, T$ **do**
    Compute $\hat{\theta}_k(t)$ for all $k$
    $\mathcal{L}(t) \leftarrow$ top-$L$ ordered arms by decreasing $\hat{\theta}_k(t)$
    $A_l(t) \leftarrow \mathcal{L}_l(t)$ for each position $l < L$
    $\mathcal{B}(t) \leftarrow \{k | k \notin \mathcal{L}(t), U_k(t, (1 + \epsilon) \log(T)) \geq \hat{\theta}_{\mathcal{L}_L(t)}(t)$
    **if** $\mathcal{B}(t) = \emptyset$ **then**
      $A_L(t) \leftarrow \mathcal{L}_L(t)$
    **else**
      With probability $1/2$, select $A_L(t)$ uniformly at random from $\mathcal{B}(t)$, else $A_L(t) \leftarrow \mathcal{L}_L(t)$
    **end if**
    Play action $A(t)$ and observe feedback $Z(t)$; Update $N_{k,l}(t + 1)$ and $S_{k,l}(t + 1)$.
  **end for**

---

The $U_k(t, \delta)$ index defined in (7) aggregates observations from all positions – as in PBM-UCB – but allows to build tighter confidence regions as shown by the next proposition proved in Appendix E.

**Proposition 10.** *For all* $\delta \geq L + 1$,

$$\mathbb{P}(U_k(t, \delta) < \theta_k) \leq e^{L+1} \left( \frac{\lceil \delta \log(t) \rceil \delta}{L} \right)^L e^{-\delta}.$$

We may now state the main result of this section that provides an upper bound on the regret of PBM-PIE.

**Theorem 11.** *Using* PBM-PIE *with* $\delta = (1 + \epsilon) \log(t)$ *and* $\epsilon > 0$, *for any* $\eta < \min_{k < K}(\theta_k - \theta_{k+1})/2$, *there exist problem-dependent constants* $C_1(\eta), C_2(\epsilon, \eta), C_3(\epsilon)$ *and* $\beta(\epsilon, \eta)$ *such that*

$$\mathbb{E}[R(T)] \leq (1 + \epsilon)^2 \log(T) \sum_{k=L+1}^{K} \frac{\kappa_L(\theta_L - \theta_k)}{d(\kappa_L \theta_k, \kappa_L(\theta_L - \eta))} + C_1(\eta) + \frac{C_2(\epsilon, \eta)}{T^{\beta(\epsilon, \eta)}} + C_3(\epsilon).$$

The proof of this result is provided in Appendix E. Comparing to the expression in (6), Theorem 11 shows that PBM-PIE reaches asymptotically optimal performance when the optimal exploring

position is indeed located at index $L$. In other case, there is a gap that is caused by the fact the exploring position is fixed beforehand and not adapted from the data.

We conclude this section by a quick description of two other algorithms that will be used in the experimental section to benchmark our results.

**Ranked Bandits (RBA-KL-UCB)** The state-of-the-art algorithm for the sequential "learning to rank" problem was proposed by [18]. It runs one bandit algorithm per position, each one being entitled to choose the best suited arm at its rank. The underlying bandit algorithm that runs in each position is left to the choice of the user, the better the policy the lower the regret can be. If the bandit algorithm at position $l$ selects an arm already chosen at a higher position, it receives a reward of zero. Consequently, the bandit algorithm operating at position $l$ tends to focus on the estimation of $l$-th best arm. In the next section, we use as benchmark the Ranked Bandits strategy using the KL-UCB algorithm [10] as the per-position bandit.

**PBM-TS** The observations $Z_l(t)$ are censored Bernoulli which results in a posterior that does not belong to a standard family of distribution. [13] suggest a version of Thompson Sampling called "Bias Corrected Multiple Play TS" (or BC-MP-TS) that approximates the true posterior by a Beta distribution. We observed in experiments that for parameter values close to one, this algorithm does not explore enough. In Figure 1(a), we show this phenomenon for $\theta = (0.95, 0.85, 0.75, 0.65, 0.55)$. The true posterior for the parameter $\theta_k$ at time $t$ may be written as a product of truncated scaled beta distributions

$$\pi_t(\theta_k) \propto \prod_l \theta_k^{\alpha_{k,l}(t)}(1 - \kappa_l\theta_k)^{\beta_{k,l}(t)},$$

where $\alpha_{k,l}(t) = S_{k,l}(t)$ and $\beta_{k,l}(t) = N_{k,l}(t) - S_{k,l}(t)$. To draw from this exact posterior, we use rejection sampling with proposal distribution $\text{Beta}(\alpha_{k,m}(t), \beta_{k,m}(t))/\kappa_m$, where $m = \arg\max_{1 \le l \le L}(\alpha_{k,l}(t) + \beta_{k,l}(t))$.

## 5 Experiments

### 5.1 Simulations

In order to evaluate our strategies, a simple problem is considered in which $K = 5$, $L = 3$, $\kappa = (0.9, 0.6, 0.3)$ and $\theta = (0.45, 0.35, 0.25, 0.15, 0.05)$. The arm expectations are chosen such that the asymptotic behavior can be observed after reasonable time horizon. All results are averaged based on $10,000$ independent runs of the algorithm. We present the results in Figure 1(b) where PBM-UCB, PBM-PIE and PBM-TS are compared to RBA-KL-UCB. The performance of PBM-PIE and PBM-TS are comparable, the latter even being under the lower bound (it is a common observation, e.g. see [13], and is due to the asymptotic nature of the lower bound). The curves confirm our analysis

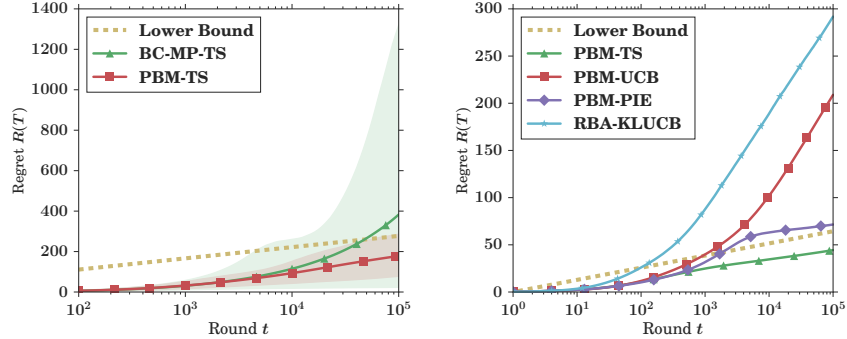

(a) Average regret of PBM-TS and BC-MP-TS compared for high parameters. Shaded areas: first and last deciles.

(b) Average regret of various algorithms on synthetic data under the PBM.

**Figure 1:** Simulation results for the suggested strategies.

| #ads ($K$) | #records | min $\theta$ | max $\theta$ |
|---|---|---|---|
| 5 | $216,565$ | $0.016$ | $0.077$ |
| 5 | $68,179$ | $0.031$ | $0.050$ |
| 6 | $435,951$ | $0.025$ | $0.067$ |
| 6 | $110,071$ | $0.023$ | $0.069$ |
| 6 | $147,214$ | $0.004$ | $0.148$ |
| 8 | $122,218$ | $0.108$ | $0.146$ |
| 11 | $1,228,004$ | $0.022$ | $0.149$ |
| 11 | $391,951$ | $0.022$ | $0.084$ |

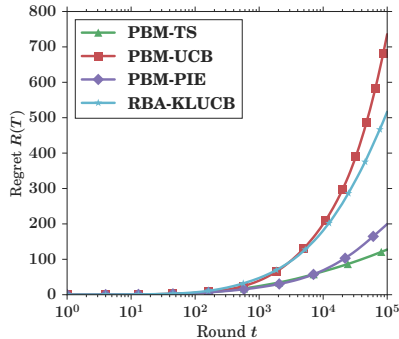

**Table 1:** Statistics on the queries: each line corresponds to the sub-dataset associated with a query.

**Figure 2:** Performance of the proposed algorithms under the PBM on real data.

for PBM-PIE and lets us conjecture that the true Thompson Sampling policy might be asymptotically optimal. As expected, PBM-PIE shows asymptotically optimal performance, matching the lower bound after a large enough horizon.

### 5.2 Real data experiments: search advertising

The dataset was provided for KDD Cup 2012 track 2[1] and involves session logs of soso.com, a search engine owned by Tencent. It consists of ads that were inserted among search results. Each of the $150M$ lines from the log contains the user ID, the query typed, an ad, a position (1, 2 or 3) at which it was displayed and a binary reward (click/no-click). First, for every query, we excluded ads that were not displayed at least $1,000$ times at every position. We also filtered queries that had less than $5$ ads satisfying the previous constraints. As a result, we obtained $8$ queries with at least $5$ and up to $11$ ads. For each query $q$, we computed the matrix of the average click-through rates (CTR): $M_q \in \mathbb{R}^{K \times L}$, where $K$ is the number of ads for the query $q$ and $L = 3$ the number of positions. It is noticeable that the SVD of each $M_q$ matrix has a highly dominating first singular value, therefore validating the low-rank assumption underlying in the PBM. In order to estimate the parameters of the problem, we used the EM algorithm suggested by [5, 9]. Table 1 reports some statistics about the bandit models reconstructed for each query: number of arms $K$, amount of data used to compute the parameters, minimum and maximum values of the $\theta$'s for each model.

We conducted a series of $2,000$ simulations over this dataset. At the beginning of each run, a query was randomly selected together with corresponding probabilities of scanning positions and arm expectations. Even if rewards were still simulated, this scenario is more realistic since the values of the parameters were extracted from a real-world dataset. We show results for the different algorithms in Figure 2. It is remarkable that RBA-KL-UCB performs slightly better than PBM-UCB. One can imagine that PBM-UCB does not benefit enough from position aggregations – only $3$ positions are considered – to beat RBA-KL-UCB. Both of them are outperformed by PBM-TS and PBM-PIE.

## Conclusion

This work provides the first analysis of the PBM in an online context. The proof scheme used to obtain the lower bound on the regret is interesting on its own, as it can be generalized to various other settings. The tightness of the lower bound is validated by our analysis of PBM-PIE but it would be an interesting future contribution to provide such guarantees for more straightforward algorithms such as PBM-TS or a 'PBM-KLUCB' using the confidence regions of PBM-PIE. In practice, the algorithms are robust to small variations of the values of the $(\kappa_l)_l$, but it would be preferable to obtain some control over the regret under uncertainty on these examination parameters.

## Acknowledgements

This work was partially supported by the French research project ALICIA (grant ANR-13-CORD-0020) and by the Machine Learning for Big Data Chair at Télécom ParisTech.

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
