[Supplementary Material · nips_appendix.pdf]

# A    Properties of $\hat{\theta}_k(t)$ (Section 2)

Conditionnally to the actions $A(1)$ up to $A(t-1)$, the log-likelihood of the observations $Z(1),\ldots,Z(t-1)$ may be written as

$$\sum_{s=}^{t-1}\sum_{k=1}^{K}\sum_{l=1}^{L}\mathbb{1}\{A_l(t)=k\}\left[Z_l(t)\log(\kappa_l\theta_k)+(1-Z_l(t))\log(1-\kappa_l\theta_k)\right]$$

$$=\sum_{k=1}^{K}\sum_{l=1}^{L}S_{k,l}(t)\log(\kappa_l\theta_k)+(N_{k,l}(t)-S_{k,l}(t))\log(1-\kappa_l\theta_k).$$

Differenciating twice with respect to $\theta_k$ and taking the expectation of $(S_{k,l}(t))_l$, contional to $A(1),\ldots,A(t-1)$, yields the expression of $I(\theta_k)$ given in Section 2.

# B    Proof of Theorem 4

## B.1    Proof of Lemma 2

Under the PBM, the conditional expectation of the log-likelihood ratio defined in (4) writes

$$\mathbb{E}_\theta[\ell(t)|A(1),\ldots,A(t)]=\mathbb{E}_\theta\left[\sum_{s=1}^{t}\sum_{a\in\mathcal{A}}\mathbb{1}\{A(s)=a\}\sum_{l=1}^{L}\log\frac{p_{a_l}(X_l(s)Y_l(s);\theta)}{p_{a_l}(X_l(s)Y_l(s);\lambda)}\;\middle|\;A(1),\ldots,A(t)\right]$$

$$=\sum_{s=1}^{t}\sum_{a\in\mathcal{A}}\mathbb{1}\{A(s)=a\}\sum_{l=1}^{L}\mathbb{E}\left[\log\frac{p_{a_l}(X_l(s)Y_l(s);\theta)}{p_{a_l}(X_l(s)Y_l(s);\lambda)}\;\middle|\;A(s)=a\right]$$

$$=\sum_{a\in\mathcal{A}}N_a(t)\sum_{l=1}^{L}\sum_{k=1}^{K}\mathbb{1}\{a_l=k\}d(\kappa_l\theta_k,\kappa_l\lambda_k)$$

$$=\sum_{a\in\mathcal{A}}N_a(t)I_a(\theta,\lambda),$$

using the notation $I_a(\theta,\lambda)=\sum_{l=1}^{L}\sum_{k=1}^{K}\mathbb{1}\{a_l=k\}d(\kappa_l\theta_k,\kappa_l\lambda_k)$. $\qquad\square$

## B.2    Details on the proof of Proposition 3

**Lemma 12.** *Let $\theta=(\theta_1,\ldots,\theta_K)$ and $\lambda=(\lambda_1,\ldots,\lambda_K)$ be two bandit models such that the distributions of all arms in $\theta$ and $\lambda$ are mutually absolutely continuous. Let $\sigma$ be a stopping time with respect to $(\mathcal{F}_t)$ such that $(\sigma<+\infty)$ a.s. under both models. Let $\mathcal{E}\in\mathcal{F}_\sigma$ be an event such that $0<\mathbb{P}_\theta(\mathcal{E})<1$. Then one has*

$$\sum_{a\in\mathcal{A}}I_a(\theta,\lambda)\mathbb{E}_\theta[N_a(\sigma)]\geq d(\mathbb{P}_\theta(\mathcal{E}),\mathbb{P}_\lambda(\mathcal{E})),$$

*where $I_a(\theta,\lambda)$ is the conditional expectation of the log-likelihood ratio for the model of interest.*

The proof of this lemma directly follows from the above expressions of the log-likelihood ratio and from the proof of Lemma 1 in Appendix A.1 of [12].

We simply recall the following technical lemma for completeness.

**Lemma 13.** *Let $\sigma$ be any stopping time with respect to $(\mathcal{F}_t)$. For every event $A\in\mathcal{F}_\sigma$,*

$$\mathbb{P}_\lambda(A)=\mathbb{E}_\theta[\mathbb{1}\{A\}\exp(-\ell(\sigma))].$$

A full proof of Lemma 13 can be found in the Appendix A.3 of [12] (proof of Lemma 15).

## B.3 Lower bound proof (Theorem 4)

In order to prove the simplified lower bound of Theorem 4 we basically have two arguments:

1. a lower bound on $f(\theta)$ can be obtained by enlarging the feasible set, that is by relaxing some constraints;

2. Lemma 15 can be used to lower bound the objective function of the problem.

The constant $f(\theta)$ is defined by

$$f(\theta) = \inf_{c \succeq 0} \sum_{a \neq a^*(\theta)} \Delta_a(\theta) c_a \tag{8}$$

$$s.t \quad \inf_{\lambda \in B(\theta)} \sum_{a \in \mathcal{A}} I_a(\theta, \lambda) c_a \geq 1. \tag{9}$$

We begin by relaxing some constraints: we only allow the change of measure $\lambda$ to belong to the sets $B_k(\theta) := \{\lambda \in \Theta | \forall j \neq k, \theta_j = \lambda_j \text{ and } \mu^*(\theta) < \mu^*(\lambda)\}$ defined in Section 3:

$$f(\theta) = \inf_{c \succeq 0} \sum_{a \neq a^*(\theta)} \Delta_a(\theta) c_a \tag{10}$$

$$s.t \quad \forall k \notin a^*(\theta), \ \forall \lambda \in B_k(\theta), \sum_{a \in \mathcal{A}} I_a(\theta, \lambda) c_a \geq 1. \tag{11}$$

The $K - L$ constraints (11) only let one parameter move and must be true for any value satisfying the definition of the corresponding set $B_k(\theta)$. In practice, for each $k$, the parameter $\lambda_k$ must be set to at least $\theta_L$. Consequently, these constraints may then be rewritten

$$f(\theta) = \inf_{c \succeq 0} \sum_{a \neq a^*(\theta)} \Delta_a(\theta) c_a \tag{12}$$

$$s.t \ \forall k \notin a^*(\theta), \ \sum_{a \neq a^*(\theta)} c_a \sum_{l=1}^{L} \mathbb{1}\{a_l = k\} d(\kappa_l \theta_k, \kappa_l \theta_L) \geq 1. \tag{13}$$

Proposition 14 tells us that coefficients $c_a$ are all zeros except for actions $a \in \mathcal{A}$ which can be written $a = v_{k,l_k}$ where $l_k = \arg\min_{l \leq L} \frac{\Delta_{v_{k,l}}(\theta)}{d(\kappa_l \theta_k, \kappa_l \theta_L)}$. Thus, we obtain the desired lower bound by rewriting (12) as

$$f(\theta) \geq \sum_{k=L+1}^{K} \min_{l \in \{1, \ldots, L\}} \frac{\Delta_{v_{k,l}}(\theta)}{d(\kappa_l \theta_k, \kappa_l \theta_L)}.$$

$\square$

**Proposition 14.** *Let $c = \{c_a : a \neq a^*\}$ be a solution of the linear problem (LP) in Theorem 4. Coefficients are all zeros except for actions $a$ which can be written as $a = (1, \ldots, l_k-1, k, l_k, \ldots, L-1) := v_{k,l_k}$ where $k > L$ and $l_k = \arg\min_{l \leq L} \frac{\Delta_{v_{k,l}}(\theta)}{d(\kappa_l \theta_k, \kappa_l \theta_L)}$.*

*Proof.* We denote by $\pi_k(a)$ the position of item $k \in \{1, \ldots, K\}$ in action $a$ (0 if $k \notin a$). Let $l_k$ be the optimal position of item $k > L$ for exploration: $l_k = \arg\min_{l \leq L} \frac{\Delta_{v_{k,l}}(\theta)}{d(\kappa_l \theta_k, \kappa_l \theta_L)}$. Following [6], we show by contradiction that $c_a > 0$ implies that $a$ can be written $v_{k,l_k}$ for a well chosen $k > L$. Let $\alpha \neq a^*$ be a suboptimal action such that $\forall k > L, \alpha \neq v_{k,l_k}$ and $c_\alpha > 0$. We need to show a contradiction. Let us introduce a new set of coefficients $c'$ defined as follows, for any $a \neq a^*$:

$$c'_a = \begin{cases} 0 & \text{if } a = \alpha \\ c_a + \frac{d(\kappa_{\pi_k(\alpha)} \theta_k, \kappa_{\pi_k(\alpha)} \theta_L)}{d(\kappa_{l_k} \theta_k, \kappa_{l_k} \theta_L)} c_\alpha & \text{if } \exists k > L \text{ s.t. } a = v_{k,l_k} \text{ and } k \in \alpha \\ c_a & \text{otherwise.} \end{cases}$$

According to Lemma 15, these coefficients satisfy the constraints of the LP. We now show that these new coefficients yield a strictly lower value to the optimization problem:

$$c(\theta) - c'(\theta) = c_\alpha \Delta_\alpha(\theta) - \sum_{k > L : k \in \alpha} \frac{d(\kappa_{\pi_k(\alpha)} \theta_k, \kappa_{\pi_k(\alpha)} \theta_L)}{d(\kappa_{l_k} \theta_k, \kappa_{l_k} \theta_L)} c_\alpha \Delta_{v_{k,l_k}}(\theta)$$

$$> c_\alpha \left( \sum_{k > L : k \in \alpha} \Delta_{v_{k, \pi_k(\alpha)}}(\theta) - \sum_{k > L : k \in \alpha} \frac{d(\kappa_{\pi_k(\alpha)} \theta_k, \kappa_{\pi_k(\alpha)} \theta_L)}{d(\kappa_{l_k} \theta_k, \kappa_{l_k} \theta_L)} \Delta_{v_{k,l_k}}(\theta) \right). \quad (14)$$

The strict inequality (14) is shown in Lemma 16. Let $k > L$ be one of the suboptimal arms in $\alpha$. By definition of $l_k$, the corresponding term of the sum in equation (14) is positive. Thus, we have that $c(\theta) > c'(\theta)$ and, hence, by contradiction, we showed that $c_a > 0$ iff $a$ can be written $a = v_{k,l_k}$ for some $k > L$. $\qquad\square$

**Lemma 15.** *Let $c$ be a vector of coefficients that satisfy constraints (13) of the optimization problem. Then, coefficients $c'$ as defined in Proposition 14 also satisfy the constraints:*

$$\forall k \notin a^*(\theta), \quad \sum_{a \neq a^*(\theta)} c'_a \sum_{l=1}^{L} \mathbb{1}\{a_l = k\} d(\kappa_l \theta_k, \kappa_l \theta_L) \geq 1.$$

*Proof.* We use the same $\alpha$ as introduced in Proposition 14. Let us fix $k \notin a^*(\theta)$. Let us define

$$L(c) = \sum_{a \neq a^*(\theta)} c_a \sum_{l=1}^{L} \mathbb{1}\{a_l = k\} d(\kappa_l \theta_k, \kappa_l \theta_L).$$

We have

$$L(c') - L(c) = -c_\alpha \sum_{l=1}^{L} \mathbb{1}\{\alpha_l = k\} d(\kappa_l \theta_k, \kappa_l \theta_L) + \sum_{l : \alpha_l > L} \frac{d(\kappa_l \theta_k, \kappa_l \theta_L)}{d(\kappa_{l_k} \theta_k, \kappa_{l_k} \theta_L)} c_\alpha$$
$$\times \mathbb{1}\{\alpha_l = k\} d(\kappa_{l_k} \theta_k, \kappa_{l_k} \theta_L).$$

If $k \notin \alpha$, clearly, $L(c') - L(c) = 0$. Else, $k \in \alpha$ and we note $p$ its position in $\alpha$: $p = \pi_k(\alpha)$. We rewrite:

$$L(c') - L(c) = c_\alpha d(\kappa_p \theta_k, \kappa_p \theta_L) \left( -1 + \frac{d(\kappa_{l_k} \theta_k, \kappa_{l_k} \theta_L)}{d(\kappa_{l_k} \theta_k, \kappa_{l_k} \theta_L)} \right) = 0.$$

Thus, the coefficients $c'$ satisfy the constraints from Proposition 14. $\qquad\square$

**Lemma 16.** *Let $\alpha$ be as in the proof of Proposition 14.*

$$\Delta_\alpha(\theta) > \sum_{k > L : k \in \alpha} \Delta_{v_{k, \pi_k(\alpha)}}(\theta).$$

*Proof.* Let $k_1, \ldots, k_p$ be the suboptimal arms in $\alpha$ by increasing position. Let $v(\alpha)$ be the action in $\mathcal{A}$ with lower regret such that it contains all the suboptimal arms of $\alpha$ in the same positions. Thus, $v(\alpha) = (1, \ldots, \pi_{k_1}(\alpha) - 1, k_1, \pi_{k_1}(\alpha), \ldots, \pi_{k_2}(\alpha) - 2, k_2, \pi_{k_2}(\alpha) - 1, \ldots, L - p)$. By definition, one has that $\Delta_\alpha(\theta) \geq \Delta_{v(\alpha)}(\theta)$. In the following, we show that $\Delta_{v(\alpha)}(\theta) \geq \sum_{k > L : k \in \alpha} \Delta_{v_{k, \pi_k(\alpha)}}(\theta)$ for $p = 2$ (that is to say $\alpha$ contains 2 suboptimal arms $k_1$ and $k_2$).

For the sake of readability, we write $\pi_i$ instead of $\pi_{k_i}(\alpha)$ in the following.

$$\Delta_{v(\alpha)}(\theta) = \sum_{l=1}^{L} \kappa_l (\theta_l - \theta_{(v_{k_1, \pi_1})_l}) + \sum_{l=1}^{L} \kappa_l (\theta_{(v_{k_1, \pi_1})_l} - \theta_{v(\alpha)_l})$$
$$= \Delta_{v_{k_1, \pi_1}}(\theta) + [\kappa_{\pi_2} \theta_{\pi_2 - 1} + \ldots + \kappa_L \theta_{L-1}] - [\kappa_{\pi_2} \theta_{k_2} + \kappa_{\pi_2 + 1} \theta_{\pi_2 - 1} + \ldots + \kappa_L \theta_{L-2}]$$
$$= \Delta_{v_{k_1, \pi_1}}(\theta) + \Delta_{v_{k_2, \pi_2}}(\theta) + [\kappa_{\pi_2}(\theta_{\pi_2 - 1} - \theta_{\pi_2}) + \ldots + \kappa_L(\theta_{L-1} - \theta_L)] -$$
$$[\kappa_{\pi_2 + 1}(\theta_{\pi_2 - 1} - \theta_{\pi_2}) + \ldots + \kappa_L(\theta_{L-2} - \theta_{L-1})]$$
$$= \Delta_{v_{k_1, \pi_1}}(\theta) + \Delta_{v_{k_2, \pi_2}}(\theta) + \mathcal{R}(\theta).$$

Thus, one has to show that $\mathcal{R}(\theta) = \kappa_{\pi_2}(\theta_{\pi_2-1} - \theta_{\pi_2}) + \kappa_{\pi_2+1}(2\theta_{\pi_2} - \theta_{\pi_2-1} - \theta_{\pi_2+1}) + \ldots + \kappa_L(2\theta_{L-1} - \theta_{L-2} - \theta_L) > 0$. In fact, using that $\kappa_l \geq \kappa_{l+1}$ for all $l < L$, we have

$$
\begin{aligned}
\mathcal{R}(\theta) &\geq \kappa_{\pi_2+1}(\theta_{\pi_2-1} - \theta_{\pi_2} + 2\theta_{\pi_2} - \theta_{\pi_2-1} - \theta_{\pi_2+1}) + \ldots + \kappa_L(2\theta_{L-1} - \theta_{L-2} - \theta_L) \\
&\geq \kappa_{\pi_2+2}(\theta_{\pi_2+1} - \theta_{\pi_2+2}) + \ldots + \kappa_L(2\theta_{L-1} - \theta_{L-2} - \theta_L) \\
&\geq \ldots \\
&\geq \kappa_L(\theta_{L-1} - \theta_L) \\
&> 0.
\end{aligned}
$$

$\square$

## C  Proof of Proposition 8

In this section, we fix an arm $k \in \{1, \ldots, K\}$ and obtain an upper confidence bound for the estimator $\hat{\theta}_k(t) := S_k(t)/\tilde{N}_k(t)$. Let $\tau_i$ be the instant of the $i$-th draw of arm $k$ (the $\tau_i$ are stopping times w.r.t. $\mathcal{F}_t$). We introduce the centered sequence of successive observations from arm $k$

$$
\bar{Z}_{k,i} = \sum_{l=1}^{L} \mathbb{1}\{A_l(\tau_i) = k\}(X_l(\tau_i)Y_l(\tau_i) - \theta_k\kappa_l). \tag{15}
$$

Introducing the filtration $\mathcal{G}_i = \mathcal{F}_{\tau_{i+1}-1}$, one has $\mathbb{E}[\bar{Z}_{k,i}|\mathcal{G}_{i-1}] = 0$, and therefore, the sequence

$$
M_{k,n} = \sum_{i=1}^{n} \bar{Z}_{k,i}
$$

is a martingale with bounded increments, w.r.t. the filtration $(\mathcal{G}_n)_n$. By construction, one has

$$
M_{k,N_k(t)} = S_k(t) - \tilde{N}_k(t)\theta_k = \tilde{N}_k(t)(\hat{\theta}_k(t) - \theta_k).
$$

We use the so-called peeling technique together with the maximal version of Azuma-Hoeffding's inequality [3]. For any $\gamma > 0$ one has

$$
\begin{aligned}
\mathbb{P}\left(M_{k,N_k(t)} < -\sqrt{N_k(t)\delta/2}\right) &\leq \sum_{i=1}^{\frac{\log(t)}{\log(1+\gamma)}} \mathbb{P}\left(M_{k,N_k(t)} < -\sqrt{N_k(t)\delta/2}, N_k(t) \in [(1+\gamma)^{i-1}, (1+\gamma)^i)\right) \\
&\leq \sum_{i=1}^{\frac{\log(t)}{\log(1+\gamma)}} \mathbb{P}\left(\exists i \in \{1, \ldots, (1+\gamma)^i\} : M_{k,i} < -\sqrt{(1+\gamma)^{i-1}\delta/2}\right) \\
&\leq \sum_{i=1}^{\frac{\log(t)}{\log(1+\gamma)}} \exp\left(-\frac{\delta(1+\gamma)^{i-1}}{(1+\gamma)^i}\right) = \frac{\log(t)}{\log(1+\gamma)} \exp\left(-\frac{\delta}{(1+\gamma)}\right).
\end{aligned}
$$

Choosing $\gamma = 1/(\delta - 1)$, gives

$$
\mathbb{P}\left(\hat{\theta}_k(t) - \theta_k < -\frac{\sqrt{N_k(t)\delta/2}}{\tilde{N}_k(t)}\right) \leq \delta e \log(t) e^{-\delta}.
$$

## D  Regret analysis for PBM-UCB (Theorem 9)

We proceed as Kveton et al. (2015) [15]. We start by considering separately rounds when one of the confidence intervals is violated. We denote by $B_{t,k} = \sqrt{N_k(t)(1+\epsilon)\log t/2}/\tilde{N}_k(t)$ the PBM-UCB exploration bonus and by $B_{t,k}^+ = \sqrt{N_k(t)(1+\epsilon)\log T/2}/\tilde{N}_k(t)$ an upper bound of this bonus (for $t \leq T$). We define the event $E_t = \{\exists k \in A(t) : |\hat{\theta}_k(t) - \theta_k| > B_{t,k}\}$. Then, the regret can be decomposed into

$$
R(T) = \sum_{t=1}^{T} \Delta_{A(t)} \mathbb{1}_{E_t} + \Delta_{A(t)} \mathbb{1}_{\bar{E}_t}.
$$

and, similarly to [15] (Appendix A.1), the first term of this sum can be bounded from above in expectation by a constant $C_0(\epsilon)$ that does not depend on $T$ using Proposition 8. So, it remains to bound the regret suffered even when confidence intervals are respected, that is the sum on the r.h.s of

$$\mathbb{E}[R(T)] < C_0(\epsilon) + \mathbb{E}[\sum_{t=1}^{T} \Delta_{A(t)} \mathbb{1}\{\bar{E}_t, \Delta_{A(t)} > 0\}].$$

It can be done using techniques from [7, 15]. We start by defining events $F_t$, $G_t$, $H_t$ in order to decompose the part of the regret at stake. Then, we show an equivalent of Lemma 2 of [15] for our case and finally we refer to the proof of Theorem 3 in Appendix A.3 of [15].

For each round $t \geq 1$, we define the set of arms $S_t = \{1 \leq l \leq L : N_{A_l(t)}(t) \leq \frac{8(1+\epsilon) \log T (\sum_{s=1}^{L} \kappa_s)^2}{\kappa_L^2 \Delta_{A(t)}^2}\}$ and the related events

- $F_t = \{\Delta_{A(t)} > 0, \Delta_{A(t)} \leq 2 \sum_{l=1}^{L} \kappa_l B_{t,A_l(t)}^+\}$;
- $G_t = \{|S_t| \geq l\}$;
- $H_t = \{|S_t| < l, \exists k \in A(t), N_k(t) \leq \frac{8(1+\epsilon) \log T (\sum_{s=1}^{l} \kappa_s)^2}{\kappa_L^2 \Delta_{A(t)}^2}\}$, where the constraint on $N_k(t)$ only differs from the first one by its numerator which is smaller than the previous one, leading to an even stronger constraint.

**Fact 17.** *According to Lemma 1 in [15], the following inequality is still valid with our own definition of $F_t$ :*

$$\sum_{t=1}^{T} \Delta_{A(t)} \mathbb{1}\{\bar{E}_t, \Delta_{A(t)} > 0\} \leq \sum_{t=1}^{T} \Delta_{A(t)} \mathbb{1}\{F_t\}.$$

*Proof.* Invoking Lemma 1 from [15] needs to be justified as our setting is quite different. Taking action $A(t)$ means that

$$\sum_{l=1}^{L} \kappa_l U_{A_l(t)}(t) \geq \sum_{l=1}^{L} \kappa_l U_l(t).$$

Under event $\bar{E}_t$, all UCB's are above the true parameter $\theta_k$ so we have

$$\sum_{l=1}^{L} \kappa_l (\theta_{A_l(t)} + 2B_{t,A_l(t)}) \geq \sum_{l=1}^{L} \kappa_l (\theta_l + B_{t,l}) \geq \sum_{l=1}^{L} \kappa_l \theta_l.$$

Rearranging the terms above and using $B_{t,l(t)} \leq B_{t,l(t)}^+$, we obtain

$$\sum_{l=1}^{L} \kappa_l B_{t,A_l(t)}^+ \geq 2 \sum_{l=1}^{L} \kappa_l B_{t,A_l(t)} \geq \Delta_{A(t)}.$$

$\square$

We now have to prove an equivalent of Lemma 2 in [7] that would allow us to split the right-hand side above in two parts. Let us show that $F_t \subset (G_t \cup H_t)$ by showing its contrapositive: if $F_t$ is true then we cannot have $(\bar{G}_t \cap \bar{H}_t)$. Assume both of these events are true. Then, we have

$$\Delta_{A(t)} \overset{F_t}{\leq} 2 \sum_{l=1}^{L} \kappa_l B_{t,A_l(t)}^+$$

$$\leq 2 \sum_{l=1}^{L} \kappa_l \sqrt{\frac{N_{A_l(t)}(t)}{\tilde{N}_{A_l(t)}(t)}} \sqrt{\frac{(1+\epsilon) \log(T)}{2\tilde{N}_{A_l(t)}(t)}}$$

$$= 2 \sum_{l=1}^{L} \kappa_l \frac{N_{A_l(t)}(t)}{\tilde{N}_{A_l(t)}(t)} \sqrt{\frac{(1+\epsilon) \log(T)}{2N_{A_l(t)}(t)}}$$

$$\leq \frac{\sqrt{2(1+\epsilon)\log T}}{\kappa_L} \sum_{l=1}^{L} \frac{\kappa_l}{\sqrt{N_{A_l(t)}(t)}}$$

$$= \frac{\sqrt{2(1+\epsilon)\log T}}{\kappa_L} \left( \sum_{l \notin S_t} \frac{\kappa_l}{\sqrt{N_{A_l(t)}(t)}} + \sum_{l \in S_t} \frac{\kappa_l}{\sqrt{N_{A_l(t)}(t)}} \right)$$

$$\overset{(\bar{G}_t \cap \bar{H}_t)}{<} \frac{\sqrt{2(1+\epsilon)\log T}}{\kappa_L} \frac{\kappa_L \Delta_{A(t)}}{2\sqrt{2(1+\epsilon)\log T}} \left( \frac{\sum_{l \notin S_t} \kappa_l}{\sum_{s=1}^{L} \kappa_s} + \frac{\sum_{l \in S_t} \kappa_l}{\sum_{s=1}^{l} \kappa_s} \right)$$

$$\leq \Delta_{A(t)}$$

which is a contradiction. The end of the proof proceeds exactly as in the end of the proof of Theorem 6 in of [7]: events $G_t$ and $H_t$ are split into subevents corresponding to rounds where each specific suboptimal arm of the list is in $S_t$ or verifies the condition of $H_t$. We define

$$G_{k,t} = G_t \cap \{k \in A(t), \ N_k(t) \leq \frac{8(1+\epsilon)\log T \left(\sum_{s=1}^{L} \kappa_s\right)^2}{\kappa_L^2 \Delta_{A(t)}^2} \},$$

$$H_{k,t} = H_t \cap \{k \in A(t), \ N_k(t) \leq \frac{8(1+\epsilon)\log T \left(\sum_{s=1}^{l} \kappa_s\right)^2}{\kappa_L^2 \Delta_{A(t)}^2} \}.$$

The way we defined these subevents allows to write the two following bounds :

$$\sum_{k=1}^{K} \mathbb{1}\{G_{k,t}\} = \mathbb{1}\{G_t\} \sum_{k=1}^{K} \mathbb{1}\{k \in S_t\} \geq l\mathbb{1}\{G_t\}$$

so $\mathbb{1}\{G_t\} \leq \sum_k \mathbb{1}\{G_{k,t}\}/l$. And,

$$\mathbb{1}\{H_t\} \leq \sum_{k=1}^{K} \mathbb{1}\{H_{k,t}\}.$$

We can now bound the regret using these two results:

$$\sum_{t=1}^{T} \Delta_{A(t)}(\mathbb{1}\{G_t\} + \mathbb{1}\{H_t\}) \leq \sum_{t=1}^{T}\sum_{k=1}^{K} \frac{\Delta_{A(t)}}{l}\mathbb{1}\{G_{k,t}\} + \sum_{t=1}^{T}\sum_{k=1}^{K} \Delta_{A(t)}\mathbb{1}\{H_{k,t}\}$$

$$= \sum_{t=1}^{T}\sum_{k=1}^{K} \frac{\Delta_{A(t)}}{l}\mathbb{1}\{G_{k,t}, A(t) \neq a^*\} + \sum_{t=1}^{T}\sum_{k=1}^{K} \Delta_{A(t)}\mathbb{1}\{H_{k,t}, A(t) \neq a^*\}.$$

For each arm $k$, there is a finite number $C_k := |\mathcal{A}_k|$ of actions in $\mathcal{A}$ containing $k$; we order them such that the corresponding gaps are in decreasing order $\Delta_{k,1} \geq \ldots \geq \Delta_{k,C_k} > 0$. So we decompose each sum above on the different actions $A(t)$ possible:

$$\ldots \leq \sum_{t=1}^{T}\sum_{k=1}^{K}\sum_{a \in \mathcal{A}_k} \frac{\Delta_{k,a}}{l}\mathbb{1}\{G_{k,t}, A(t) = a\} + \sum_{t=1}^{T}\sum_{k=1}^{K}\sum_{a \in \mathcal{A}_k} \Delta_{k,a}\mathbb{1}\{H_{k,t}, A(t) = a\}.$$

The two sums on the right hand side look alike. For arm $k$ fixed, events $G_{k,t}$ and $H_{k,t}$ imply almost the same condition on $N_k(t)$, only $H_{k,t}$ is stronger because the bounding term is smaller. We now rely on a technical result by [7] that allows to bound each sum.

**Lemma 18.** *([7], Lemma 2 in Appendix B.4) Let $k$ be a fixed item and $|\mathcal{A}_k| \geq 1$, $C > 0$, we have*

$$\sum_{t=1}^{T} \sum_{a \in \mathcal{A}_k} \mathbb{1}\{k \in A(t), \ N_k(t) \leq C/\Delta_{k,a}^2, \ A(t) = a\}\Delta_{k,a} \leq \frac{2C}{\Delta_{min,k}}$$

*where $\Delta_{min,k}$ is the smallest gap among all suboptimal actions containing arm k. In particular, when $k \notin a^*$ the smallest gap is $\Delta_{min,k} = \kappa_L(\theta_L - \theta_k)$. While, when $k \in a^*$ it is less obvious what the minimal gap is, however it corresponds the second best action $A_2$ containing only optimal arms: $\Delta_{min,k} = \Delta_{A_2}$.*

So, bounding each sum with the above lemma, we obtain

$$\sum_{t=1}^{T} \Delta_{A(t)}(\mathbb{1}\{G_t\}+\mathbb{1}\{H_t\}) \le \frac{16(1+\epsilon)\log T}{\kappa_L^2} \underbrace{\left(\frac{\left(\sum_{s=1}^{L}\kappa_s\right)^2}{l} + \left(\sum_{s=1}^{l}\kappa_s\right)^2\right)}_{C(l;\kappa)} \left(\frac{L}{\Delta_{A_2}} + \sum_{k\notin a^*}\frac{1}{\kappa_L(\theta_L-\theta_k)}\right).$$

This bound can be optimized by minimizing $C(l;\kappa)$ over $l$.

## E Regret analysis for PBM-PIE (Theorem 11)

The proof follows the decomposition of [6]. For all $t \ge 1$, we denote $f(t,\epsilon) = (1+\epsilon)\log t$.

### E.1 Controlling leaders and estimations

Define $\eta_0 = \min_{k\in\{1,\dots,L-1\}}(\theta_k - \theta_{k+1})/2$ and let $\eta < \eta_0$. We define the following set of rounds

$$A = \{t \ge 1 : \mathcal{L}(t) \ne (1,\dots,L)\}.$$

Our goal is to upper bound the expected size of $A$. Let us introduce the following sets of rounds:

$$\begin{aligned}
B &= \{t \ge 1 : \exists k \in \mathcal{L}(t), |\hat{\theta}_k(t) - \theta_k| \ge \eta\}, \\
C &= \{t \ge 1 : \exists k \le L, U_k(t) \le \theta_k\}, \\
D &= \{t \ge 1 : t \in A \setminus (B \cup C), \exists k \le L, k \notin \mathcal{L}(t), |\hat{\theta}_k(t) - \theta_k| \ge \eta\}.
\end{aligned}$$

We first show that $A \subset (B \cup C \cup D)$. Let $t \in A \setminus (B \cup C)$. Let $k, k' \in \mathcal{L}(t)$ such that $k < k'$. Since $t \notin B$, we have that $|\hat{\theta}_k(t) - \theta_k| \le \eta$ and $|\hat{\theta}_{k'}(t) - \theta_{k'}| \le \eta$. Since $\eta \le (\theta_k - \theta_{k'})/2$, we conclude that $\hat{\theta}_k(t) \ge \hat{\theta}_{k'}(t)$. This proves that $(\mathcal{L}_1(t),\dots,\mathcal{L}_L(t))$ is an increasing sequence. We have that $\mathcal{L}_L(t) > L$ otherwise $\mathcal{L}(t) = (1,\dots,L)$ which is a contradiction because $t \in A$. Since $\mathcal{L}_L(t) > L$, there exists $k \le L$ such that $k \notin \mathcal{L}(t)$. We show by contradiction that $|\hat{\theta}_k(t) - \theta_k| \ge \eta$. Assume that $|\hat{\theta}_k(t) - \theta_k| \le \eta$. We also have that $\hat{\theta}_{\mathcal{L}_L(t)}(t) - \theta_{\mathcal{L}_L(t)} \le \eta$ because $\mathcal{L}_L(t) \in \mathcal{L}(t)$ and $t \notin B$. Thus, $\hat{\theta}_k(t) > \hat{\theta}_{\mathcal{L}_L(t)}(t)$. We have a contradiction because this would imply that $k \in \mathcal{L}(t)$. Finally we have proven that if $t \in A \setminus (B \cup C)$, then $t \in D$ so $A \subset (B \cup C \cup D)$.

By a union bound, we obtain
$$\mathbb{E}[|A|] \le [|B|] + [|C|] + [|D|].$$

In the following, we upper bound each set of rounds individually.

**Controlling $\mathbb{E}[|B|]$:** We decompose $B = \bigcup_{k=1}^{K}(B_{k,1} \cup B_{k,2})$ where

$$\begin{aligned}
B_{k,1} &= \{t \ge 1 : k \in \mathcal{L}(t), \mathcal{L}_L(t) \ne k, |\hat{\theta}_k(t) - \theta_k| \ge \eta\} \\
B_{k,2} &= \{t \ge 1 : k \in \mathcal{L}(t), \mathcal{L}_L(t) = k, |\hat{\theta}_k(t) - \theta_k| \ge \eta\}
\end{aligned}$$

Let $t \in B_{k,1}$: $k \in A(t)$ so $\mathbb{E}[k \in A(t)|t \in B_{k,1}] = 1$. Furthermore, for all $t$, $\mathbb{1}\{t \in B_{k,1}\}$ is $\mathcal{F}_{t-1}$ measurable. Then we can apply Lemma 22 (with $H = B_{k,1}$ and $c = 1$).

$$\mathbb{E}[|B_{k,1}|] \le 2(2 + \kappa_L^{-2}\eta^{-2}).$$

Let $t \in B_{k,2}$: $k \in \mathcal{B}(t)$ but because of the randomization of the algorithm, $k \in A(t)$ with probability $1/2$, i.e. $\mathbb{E}[k \in A(t)|t \in B_{k,2}] \ge 1/2$. We get

$$\mathbb{E}[|B_{k,2}|] \le 4(4 + \kappa_L^{-2}\eta^{-2})$$

By union bound over $k$, we get $\mathbb{E}[|B|] \le 2K(10 + 3\kappa_L^{-2}\eta^{-2})$.

**Controlling** $\mathbb{E}[|C|]$**:**   We decompose $C = \bigcup_{k=1}^{L} C_k$ where $C_k = \{t \geq 1 : U_k(t) \leq \theta_k\}$

We first require to prove Proposition 10.

*Proof.* Theorem 2 of [17] implies that

$$\mathbb{P}\left( \sum_{l=1}^{L} N_{k,l}(t)d(\frac{S_{k,l}(t)}{N_{k,l}(t)}, \kappa_l \theta_k) \geq \delta \right) \leq e^{-\delta} \left( \frac{\lceil \delta \log(t) \rceil \delta}{L} \right)^L e^{L+1}.$$

The function $\Phi : x \to \sum_{l=1}^{L} N_{k,l}(t)d\left( \frac{S_{k,l}(t)}{N_{k,l}(t)}, \kappa_l x \right)$ is convex and non-decreasing on $[\theta_k^{min}(t), 1]$; the convexity is easily checked and $\theta_k^{min}(t)$ is defined as the minimum of this convex function. By definition, we have, either, $U_k(t, \delta) = 1$ and then $U_k(t, \delta) > \theta_k$, or, $U_k(t, \delta) < 1$ and $\Phi(U_k(t, \delta)) = \delta$, consequently

$$\mathbb{P}\left( U_k(t, \delta) < \theta_k \right) = \mathbb{P}\left( \Phi(U_k(t, \delta)) \leq \Phi(\theta_k) \right) = \mathbb{P}\left( \delta \leq \Phi(\theta_k) \right).$$

$\square$

Remember that $U_k(t) = U_k(t, (1 + \epsilon) \log(t)) = U_k(t, f(t, \epsilon))$. Thus, applying Proposition 10, we obtain for arm $k$,

$$\mathbb{E}[|C_k|] \leq \sum_{t=1}^{\infty} \mathbb{P}(U_k(t) \leq \theta_k) \leq \lceil e^{L+1} \rceil + \frac{e^{L+1}}{L^L} \sum_{t=\lceil e^{L+1} \rceil + 1}^{\infty} \frac{(2+\epsilon)^{2L}(\log t)^{3L}}{t^{1+\epsilon}} \leq C_3(\epsilon),$$

for some constant $C_3(\epsilon)$.

**Controlling** $\mathbb{E}[|D|]$**:**   Decompose $D$ as $D = \bigcup_{k=1}^{L} D_k$ where

$$D_k = \{t \geq 1 : t \in A \setminus (B \cup C), k \notin \mathcal{L}(t), |\hat{\theta}_k(t) - \theta_k| \geq \eta\}.$$

For a given $k \leq L$, $D_k$ is the set of rounds at which $k$ is not one of the leaders, and is not accurately estimated. Let $t \in D_k$. Since $k \notin \mathcal{L}(t)$, we must have $\mathcal{L}_L(t) > L$. In turn, since $t \notin B$, we have $|\hat{\theta}_{\mathcal{L}_L(t)}(t) - \theta_{\mathcal{L}_L(t)}| \leq \eta$, so that

$$\hat{\theta}_{\mathcal{L}_L(t)} \leq \theta_{\mathcal{L}_L(t)} + \eta \leq \theta_L + \eta \leq (\theta_L + \theta_{L+1})/2.$$

Furthermore, since $t \notin C$ and $1 \leq k \leq L$, we have $U_k(t) \geq \theta_k \geq \theta_L \geq (\theta_L + \theta_{L+1})/2 \geq \hat{\theta}_{\mathcal{L}_L(t)}$. This implies that $k \in \mathcal{B}(t)$ thus $\mathbb{E}[k \in A(t)|t \in D_k] \geq 1/(2K)$. We apply Lemma 22 with $H \equiv D_k$ and $c = 1/(2K)$ to get

$$\mathbb{E}[|D|] \leq \sum_{k=1}^{L} \mathbb{E}[|D_k|] \leq 4K(4K + \kappa_L^{-2}\eta^{-2}).$$

### E.2   Regret decomposition

We decompose the regret by distinguishing rounds in $A \cup B$ and other rounds. More specifically, we introduce the following sets of rounds for arm $k > L$:

$$E_k = \{t \geq 1 : t \notin (B \cup C \cup D), \mathcal{L}(t) = a^*, A(t) = v_{k,L}\}.$$

The set of instants at which a suboptimal action is selected now can be expressed as follows

$$\{t \geq 1 : A(t) \neq a^*\} \subset (B \cup C \cup D) \cup (\cup_{k=L+1} E_k).$$

Using a union bound, we obtain the upper bound

$$\mathbb{E}[R(T)] \leq \left( \sum_{l=1}^{L} \kappa_l \right) \mathbb{E}[|B \cup C \cup D|] + \sum_{k=L+1}^{K} \Delta_{v_{k,L}}(\theta)\mathbb{E}[|E_k|].$$

From previous boundaries, putting it all together, there exist $C_1(\eta)$ and $C_3(\epsilon)$, such that

$$\left( \sum_{l=1}^{L} \kappa_l \right) (\mathbb{E}[|B|] + \mathbb{E}[|C|] + \mathbb{E}[|D|]) \leq C_1(\eta) + C_3(\epsilon).$$

At this step, it suffices to bound events $E_k$ for all $k > L$.

### E.3 Bounding event $E_k$

We proceed similarly to [10]. Let us fix an arm $k > L$. Let $t \in E_k$: arm $k$ is pulled in position $L$, so by construction of the algorithm, we have that $k \in \mathcal{B}(t)$ and thus $U_k(t) \geq \hat{\theta}_{\mathcal{L}_L(t)}(t)$. We first show that this implies that $U_k(t) \geq \theta_L - \eta$. Since $t \in E_k$, we know that $\mathcal{L}_L(t) = L$, and since $t \notin B$, $|\hat{\theta}_L(t) - \theta_L| \leq \eta$. This leads to

$$U_k(t) \geq \hat{\theta}_{\mathcal{L}_L(t)}(t) = \hat{\theta}_L(t) \geq \theta_L - \eta.$$

Recall that $N_{k,L}(t)$ is the number of times arm $k$ was played in position $L$. By denoting $d^+(x,y) = \mathbb{1}\{x < y\}d(x,y)$, we have that

$$N_{k,L}(t)d^+(S_{k,L}(t)/N_{k,L}(t), \kappa_L(\theta_L - \eta)) \leq N_{k,L}(t)d^+(S_{k,L}(t)/N_{k,L}(t), \kappa_L U_k(t))$$

$$\leq \sum_{l=1}^{L} N_{k,l}(t)d^+(S_{k,l}(t)/N_{k,l}(t), \kappa_l U_k(t)) \leq f(t,\epsilon).$$

This implies that $\mathbb{1}\{t \in E_k\} \leq \mathbb{1}\{N_{k,L}(t)d^+(S_{k,L}(t)/N_{k,L}(t), \kappa_L(\theta_L - \eta)) \leq f(t,\epsilon)\}$.

**Lemma 19.** *([10], Lemma 7) Denoting by $\hat{\nu}_{k,s}^L$ the empirical mean of the first $s$ samples of $Z_{k,L}$, we have*

$$\sum_{t=1}^{T} \mathbb{1}\{A(t) = v_{k,L}, N_{k,L}(t)d^+(S_{k,L}(t)/N_{k,L}(t), \kappa_L(\theta_L - \eta)) \leq f(t,\epsilon)\}$$

$$\leq \sum_{s=1}^{T} \mathbb{1}\{sd^+(\hat{\nu}_{k,s}^L, \kappa_L(\theta_L - \eta)) \leq f(T,\epsilon)\}.$$

We apply Lemma 19 which is a direct translation of Lemma 7 from [10] to our problem. This yields

$$|E_k| \leq \sum_{s=1}^{T} \mathbb{1}\{sd^+(\hat{\nu}_{k,s}^L, \kappa_L(\theta_L - \eta)) \leq f(T,\epsilon)\}.$$

Let $\gamma > 0$. We define $K_T = \frac{(1+\gamma)f(T,\epsilon)}{d^+(\kappa_L\theta_k, \kappa_L(\theta_L-\eta))}$. We now rewrite the last inequality splitting the sum in two parts.

$$\sum_{s=1}^{T} \mathbb{P}(sd^+(\hat{\nu}_{k,s}^L, \kappa_L(\theta_L - \eta)) \leq f(T,\epsilon)) \leq K_T + \sum_{s=K_T+1}^{\infty} \mathbb{P}(K_Td^+(\hat{\nu}_{k,s}^L, \kappa_L(\theta_L - \eta)) \leq f(T,\epsilon))$$

$$\leq K_T + \sum_{s=K_T+1}^{\infty} \mathbb{P}(d^+(\hat{\nu}_{k,s}^L, \kappa_L(\theta_L - \eta)) \leq d(\kappa_L\theta_k, \kappa_L(\theta_L - \eta))/(1+\gamma))$$

$$\leq K_T + \frac{C_2(\gamma, \eta)}{T^{\beta(\gamma,\eta)}},$$

where last inequality comes from Lemma 20. Fixing $\gamma < \epsilon$, we obtain the desired result, which concludes the proof.

**Lemma 20.** *For each $\gamma > 0$, there exists $C_2(\gamma, \eta) > 0$ and $\beta(\gamma, \eta) > 0$ such that*

$$\sum_{s=K_T+1}^{\infty} \mathbb{P}\left(d^+(\hat{\nu}_{k,s}^L, \kappa_L(\theta_L - \eta)) \leq \frac{d(\kappa_L\theta_k, \kappa_L(\theta_L - \eta))}{1+\gamma}\right) \leq \frac{C_2(\gamma, \eta)}{T^{\beta(\gamma,\eta)}}.$$

*Proof.* If $d^+(\hat{\nu}_{k,s}^L, \kappa_L(\theta_L - \eta)) \leq \frac{d(\kappa_L\theta_k, \kappa_L(\theta_L-\eta))}{1+\gamma}$, then there exists some $r(\gamma, \eta) \in (\theta_k, \theta_L - \eta)$ such that $\hat{\nu}_{k,s}^L > \kappa_L r(\gamma, \eta)$ and

$$d(\kappa_L r(\gamma, \eta), \kappa_L(\theta_L - \eta)) = \frac{d(\kappa_L\theta_k, \kappa_L(\theta_L - \eta))}{1+\gamma}.$$

Hence,

$$\mathbb{P}\left(d^+(\hat{\nu}_{k,s}, \kappa_L \theta_L) < \frac{d(\kappa_L \theta_k, \kappa_L \theta_L)}{1+\gamma}\right) \leq \mathbb{P}\left(d(\hat{\nu}_{k,s}, \kappa_L \theta_k) > d(\kappa_L r(\gamma,\eta), \kappa_L \theta_k), \hat{\nu}_{k,s} > \kappa_L \theta_k\right)$$

$$\leq \mathbb{P}(\hat{\nu}_{k,s} > \kappa_L r(\gamma,\eta)) \leq \exp(-sd(\kappa_L r(\gamma,\eta), \kappa_L \theta_k)).$$

We obtain,

$$\sum_{t=K_T}^{\infty} \mathbb{P}\left(d^+(\hat{\nu}_{k,s}, \kappa_L \theta_L) < \frac{d(\kappa_L \theta_k, \kappa_L \theta_L)}{1+\gamma}\right) \leq \frac{\exp(-K_T d(\kappa_L r(\gamma,\eta), \kappa_L \theta_k))}{1 - \exp(-d(\kappa_L r(\gamma,\eta), \kappa_L \theta_k))} \leq \frac{C_2(\gamma,\eta)}{T^{\beta(\gamma,\eta)}},$$

for well chosen $C_2(\gamma,\eta)$ and $\beta(\gamma,\eta)$. $\qquad\square$

## F Lemmas

In this section, we recall two necessary concentration lemmas directly adapted from Lemma 4 and 5 in Appendix A of [6]. Although more involved from a probabilistic point of view, these results are simpler to establish than proposition 8 as their adaptation to the case of the PBM relies on a crude lower bound for $\tilde{N}_k(t)$, which is sufficient for proving Theorem 11..

**Lemma 21.** *For $k \in \{1, \ldots, K\}$ consider the martingale $M_{k,n} = \sum_{i=1}^{n} \bar{Z}_{k,i}$, where $\bar{Z}_{k,i}$ is defined in (15). Consider $\Phi$ a stopping time such that either $N_k(\Phi) \geq s$ or $\Phi = T + 1$. Then*

$$\mathbb{P}[|M_{k,N_k(\Phi)}| \geq N_k(\Phi)\eta, N_k(\Phi) \geq s] \leq 2\exp(-2s\eta^2). \tag{16}$$

*As a consequence,*

$$\mathbb{P}[|\hat{\theta}_k(\Phi) - \theta_k| \geq \eta, \Phi \leq T] \leq 2\exp(-2s\kappa_L^2\eta^2). \tag{17}$$

*Proof.* The first result is a direct application of Lemma 4 of [6] as $(Z_l(t))_t$ with $Z_l(t) = X_l(t)Y_l(t)$ is an independent sequence of $[0,1]$-valued variables.

For the second inequality, we use the fact that $\tilde{N}_k(t) \geq \kappa_L N_k(t)$. Hence,

$$\mathbb{P}[|\hat{\theta}_k(\Phi) - \theta_k| \geq \eta, \Phi \leq T] \leq \mathbb{P}\left[\frac{|M_{k,N_k(\Phi)}|}{\kappa_L N_k(\Phi)} \geq \eta, \Phi \leq T\right].$$

which is upper bounded using (16). $\qquad\square$

**Lemma 22.** *Fix $c > 0$ and $k \in \{1, \ldots, K\}$. Consider a random set of rounds $H \subset \mathbb{N}$, such that, for all $t$, $\mathbb{1}\{t \in H\}$ is $\mathcal{F}_{t-1}$ measurable and such that for all $t \in H$, $\{k \in \mathcal{B}(t)\}$ is true. Further assume, for all $t$, one has $\mathbb{E}[\mathbb{1}\{k \in A(t)\}|t \in H] \geq c > 0$. We define $\tau_s$ a stopping time such that $\sum_{t=1}^{\tau_s} \mathbb{1}\{t \in H\} \geq s$. Consider the random set $\Lambda = \{\tau_s : s \geq 1\}$. Then, for all $k$,*

$$\sum_{t \geq 0} \mathbb{P}[t \in \Lambda, |\hat{\theta}_k(t) - \theta_k| \geq \eta] \leq 2c^{-1}(2c^{-1} + \kappa_L^{-2}\eta^{-2})$$

The proof of this lemma follows that of Lemma 5 in [6] using the same lower bound for $\tilde{N}_k(t)$ as above.