[Reviews · NeurIPS 2016]

Reviewer 1

Summary

After Cascade and DCM models recently studied in the sequential setting, this papers takes the PBM model to the bandit setting. In PBM we explicitly model (but do not observe) the examination variable - where the user checked out the item. On the other hand, the authors assume the knowledge of their distributions (\kappa-s). The strongest contribution is the lower bound on the regret for this problem with a technique that could be used for similar settings. Besides that the authors give 2 algorithms for the problem with near-optimal regret guarantees. PS: The authors may be interested in the following paper: http://www.cs.columbia.edu/~blei/papers/LiangCharlinMcInerneyBlei2016.pdf

Qualitative Assessment

--- I appreciate that the authors clearly describe and reference previous results and even ideas they used. I this spirit I would expect much more comparison with [12] since it is the most related work. L55 mentions that [12] has some preliminary ideas, but then (2) considers the estimator from [12] that PBM-UCB is later based on. Q: Can you detail how much is (2), PBM-UCB, and its analysis different new from [12]? --- The authors assume that \kappa’s are know and claim that (L90) that the main statistical challenge is to deal with \theta-s. Q: How we could deal with the unknown kappa’s, that maybe not known to the advertiser from the beginning, if we do not have historical data, or if we want to have guaranteed performance and avoid using EM? --- The paper is well written and rigorous. However, some statements are rather telegraphic and I would appreciate much more explanation behind the claims. Q: Could you give intuition behind the proof from Section 3.2. Despite my remarks (to which I expect the authors to reply) I suggest accepting this paper.

Confidence in this Review

2-Confident (read it all; understood it all reasonably well)


Reviewer 2

Summary

This paper studies a variant of the multi-armed bandit problem in the multi-position setting, that is when the learner is allowed to select a list of L actions - as opposed to a single action - to be displayed to an incoming user. In the variant that is considered, the user is not assumed to scan the list of selected actions entirely, rather it examines item in position l of the list with some unknown probability \kappa_l. Now, rewards are classically defined as Bernoulli random variables with unknown mean parameter. In this novel setting that extends pervious works on ranking bandits, the authors provide a lower bound (Theorem 6) for this problem. In Section 4, two algorithms are introduced whose regrets are provided in Theorem 9 and Theorem 11, respectively. In Section 5, these algorithms are compared against two other algorithms directly inspired from the bandit literatute, both on simulated and real data sets.

Qualitative Assessment

I have read the feedback from the authors, and maintain my score. This is a strong paper, with a solid motivation, a well-grounded theoretical analysis, algorithmic contribution, and numerical experiments. The setting is very natural and makes a lot of sense, as it focuses on a key feature of recommender systems little addressed in the literature: namely the examination probability. The censored Bernoulli reward model does not reduce trivially to previous settings, and the paper is written in a clear way, with intuition given about contribution. It could be perhaps useful to provide similar intuitions in the appendix in the derivations of each result, as it is not always clear what is the role of each step/intermediate result. Question: Theorem 9: It may be useful to provide further intuition about the term C(\kappa): why it appears, exampel of scaling etc. Algorithm 1: The choice of a probability 1/2 of choosing between \cal B() and \cal L_L(t) looks a bit arbitrary. Also, one would naturally consider that if \kappa is known, a more natural choice could depend on \kappa. In Theorem 11, do you think one could improve on the "\kappa_L" factor that looks not needed when compared with the lower bound? Do you think playing on the 1/2 probability could help remove/reduce this factor? The most important critic one can make regarding the contribution is that the \kappa vector is assumed to be perfectly known. At the top of page 3, this assumption is discussed, and is essentially convincing. However, a natural question that is not addressed here is the effect of having only an approximate knowledge of \kappa to the final regret (Say, it is known up to some error epsilon in infinite or L2 norm). Another point is when \kappa depends on the user and a user is not observed a lot, or when it depends on the user class but the class is unknown and must be estimated. In these cases, one may want to be able to estimate \kappa while solving the bandit problem. This is however not discussed. Can you give some insights about the main difficulties in order to handle such a case (beyond the identification problem of the B(\kappa_l\theta_k)) model ? Minor comments: P.5, L. 182: the letter "delta" is very often used as a confidence level in [0,1]. I thus advise you use a different letter here for clarity. P.5,L.176 : "an simpler" P6., Algo 1: "\cal L_l(t)" is undefined. I guess this is the l-th component of \cal L(t). /////// Technical Quality: Good. The assumption that \kappa is perfectly may not always be a desirable assumption. Novelty/Originality: Good. Potential impact or usefulness: Very good. Clarity and presentation: Very good.

Confidence in this Review

3-Expert (read the paper in detail, know the area, quite certain of my opinion)


Reviewer 3

Summary

This paper considers a version of multi-play bandit (MP) problem where each position has a known discounting factor that is an extension of the undiscounted MP bandits [Anantharam+ '87, Komiyama+ '15]. Note that there are some existing studies such as cascade models [Kempe '08, Combes '15, Kveton'15] that take position based effects into considerations with different assumptions. They propose a regret lower bound that is based on the change-of-measure argument [Lai&Robbins '85, Graves&Lai'97], which appears to provide a tight constant on the top of log T factor. Two ucb-based algorithms (PBM-UCB, PBM-PIE) are proposed, and their regret is analyzed. The regret of PBM-PIE is asymptotically optimal (i.e., matching logarithmic factor) when positioning it into L-th slot is the best way. A version of Thompson sampling (PBM-TS) that simulate a posterior is also proposed without introducing its regret. These algorithms are empirically motivated by a click log dataset (kddcup dataset). As is consistent with some existing studies, TS-based algorithm outperforms UCB based ones.

Qualitative Assessment

The main contribution of this paper lies in the tightness of the analysis (mainly on the constant factor of logarithmic distribution-dependent regret) and three algorithms that are based on UCB and TS. Previously, (i) a tight analysis in the case without position-based discount is known [Anantharam+ '87, Komiyama+ '15, Combes+ '15]. This paper's result extends them to the case where lower position reduces the click-through-rate in a known manner. To the best of my knowledge, the regret lower bound (Thm 6) and the upper bound (Thm 11) are new. The analysis in this paper is *NOT* fully tight: there is still a gap between the lower and upper bounds on the top of log T factor when using position l \ne L is more informative per regret. They showed that an existing version of TS (i.e., BC-MP-TS, [Komiyama+'15, supplementary material] ) does not simulate a correct posterior, and implied a suboptimality of BC-MP-TS. Instead, they propose PBM-TS, which is based on the true posterior. Although they did not provide any analysis of this algorithm, PBM-TS is more close to the true spirit of TS, which might be of some value. Minor comments: Why BC-MP-TS is not appeared in Fig 1(b) and 2. I guess that the performance of PBM-TS and BC-MP-TS is very close in practice. pp 5. "fundamental tradeoff in the PBM": can it have any interpretations when it is applied to real-world systems? pp 8. In what sense the analysis is complete? There is still a gap on the optimal constant on the top of the logarithmic factor. The authors argue an empirical robustness of the algorithms to small changes of discount factors. Is there any ideas to analyze on this aspect? In practice, there are some works that address an externality (e.g., [Hummel&Mcafee WINE'14]) on these position-based models, and robustness on the discount factor is highly preferable (possibly in future work). Note: I have read the author feedback.

Confidence in this Review

3-Expert (read the paper in detail, know the area, quite certain of my opinion)


Reviewer 4

Summary

The paper analyzes multi-play bandits in a position based model. The model considers $K$ binary arms which can be displayed in $L$ slots. The model is position based in the sense that each of the $L$ positions have a probability of being examined by the user. If the position is examined, only then a stochastic reward from the arm displayed in the position is added to the reward in a particular round. It is assumed that the probabilities of being examined is known to the algorithm. In this setting the authors provide a regret lower-bound in the problem dependent setting. This lower bound is proved by a change of measure argument similar to that in [5] as referenced in the paper. The algorithm proposes two algorithms PBM-UCB and PBM-PIE for the problem at hand. The upper bound on regret for PBM-PIE is very close to optimal. Experiments on real world and synthetic data-sets have also been provided.

Qualitative Assessment

I like the paper overall. That is the reason I have pushed for 'Oral Level' in question 5. Here are the reasons for my ratings and a few suggestions to the authors : 1. The related work of the paper is very thorough. Even though I was not completely familiar to the prior work, I was able to grasp the related models that are there in the literature and how this paper is different from them. 2. It would be great if the authors include some discussion about PBM and DCM, for instance which of the models have been seen to perform better empirically and what are some pros and cons of both the models. 3. It has been assumed in the paper that the probability of evaluations are known to the algorithm. This is a slightly restrictive assumption, even though the authors suggest various references for estimating them. I am curious how the results are affected by errors in estimation of these values. 4. The presentation of the lower bound is very clear. The proof technique is not novel, however the result is new to the best of my knowledge. It is intuitively clear that $d(\kappa_l\theta_k, \kappa_l\theta_L )$ comes into picture. 5. The PBM-UCB algorithm is very natural and is based on the confidence bound on the estimator in (2). The regret guarantees are fairly standard, but again novel. 6. The PBM-PIE analysis is based on the KL-UCB algorithm and is similar to that of [5]. 7. The real world experiments are not completely "real" in the sense that after the rewards are estimated, everything is still simulated. However, this provides evidence that real world values for $\kappa_l$ and $\theta_k$ 's are suited for these algorithms. However, for a thorough experimentation, the authors are encouraged to try something like in "Unbiased offline evaluation of contextual-bandit-based news article recommendation algorithms" by Li et. al.

Confidence in this Review

2-Confident (read it all; understood it all reasonably well)


Reviewer 5

Summary

In this paper the authors discuss a method called Position-based click model (PBM) for sequential item placement taking user's feedback into account. They provide algorithms for usage of the PBM method for multiple problems and a lower bound for the model.

Qualitative Assessment

The paper spends significant effort in explaining the algorithms. But the comparison with existing methods and comparative advantage is missing. As far as I can tell, it doesn't event discuss figure 1(a) in the text. The figure is completely floating in the paper. Without such an analysis, it is difficult to ascertain what merit the paper has to a wider community.

Confidence in this Review

2-Confident (read it all; understood it all reasonably well)


Reviewer 6

Summary

In this paper, based on the Position-based click model (PBM), the authors propose an approach to exploit available information regarding the display position bias. And they provide a novel regret lower bound for this model. The experimental results show the efficiency of the proposed approach.

Qualitative Assessment

I’m not clear about the novelty and the practical importance in this work. Please give more details about these two aspects.

Confidence in this Review

2-Confident (read it all; understood it all reasonably well)